# Thermal and Stress Properties of Briquettes from Virginia Mallow Energetic Crops

**DOI:** 10.3390/ma15238458

**Published:** 2022-11-28

**Authors:** Marek Kurtyka, Magdalena Szwaja, Andrzej Piotrowski, Barbara Tora, Stanislaw Szwaja

**Affiliations:** 1Termo-Klima MK, Tartaczna 12, 40-749 Katowice, Poland; 2Faculty of Mechanical Engineering and Computer Science, Czestochowa University of Technology, Armii Krajowej 21, 42-200 Czestochowa, Poland; 3Faculty of Civil Engineering and Resource Management, AGH University of Science and Technology in Krakow, Mickiewicza 30, 30-059 Krakow, Poland

**Keywords:** thermal conductivity, compaction pressure, density, biomass, briquettes

## Abstract

The article discusses the influence of briquetting/compaction parameters. This includes the effects of pressure and temperature on material density and the thermal conductivity of biomass compacted into briquette samples. Plant biomass mainly consists of lignin and cellulose which breaks down into simple polymers at the elevated temperature of 200 °C. Hence, the compaction pressure, compaction temperature, density, and thermal conductivity of the tested material play crucial roles in the briquetting and the torrefaction process to transform it into charcoal with a high carbon content. The tests were realized for samples of raw biomass compacted under pressure in the range from 100 to 1000 bar and at two temperatures of 20 and 200 °C. The pressure of 200 bar was concluded as the most economically viable in briquetting technology in the tests conducted. The conducted research shows a relatively good log relationship between the density of the compacted briquette and the compaction pressure. Additionally, higher compaction pressure resulted in higher destructive force of the compacted material, which may affect the lower abrasion of the material. Regarding heat transfer throughout the sample, the average thermal conductivity for the compacted biomass was determined at a value of 0.048 ± 0.001 W/(K∙m). Finally, the described methodology for thermal conductivity determination has been found to be a reliable tool, therefore it can be proposed for other applications.

## 1. Introduction

This research considers the durability and thermal conductivity of biomass densified into briquettes, which is particularly important when using them in individual heating systems, e.g., heating boilers in domestic central heating installations. Therefore, the search for the most economically advantageous technologies and optimal parameters for compacting biomass into briquettes/pellets is a well-known research topic. The thermal and stress parameters of the compacted biomass should be of interest in both modeling and experimental analysis of biomass densified into briquettes and further into charcoal during the torrefaction process. However, the first studies regarding physical stress characterization of densified biomass were undertaken by Liu et al. [1]. Their research pointed to several important thermal and stress parameters of compacted biomass for its processing and applications in building construction. Literature in this field has been focused on the subject of searching for information on thermal conductivity and strength of the briquette/pellet materials. As indicated above, strength tests were carried out for briquettes/pellets, even though it is not a material whose strength properties will later be used as it is not used, for example, in building or construction material. In this regard, work by Gorzelany et al. [2] deserves attention, in which the compressive strength properties and durability assessment of pellets with a diameter of six and eight millimeters made of sawdust were discussed. They achieved compressive stresses from 5.4 to 7.9 MPa, and from 10.9 to 11.4 MPa for pellets with a diameter of six and eight millimeters, respectively. The authors also determined Young’s modulus for the tested samples, which ranged between 82.27 MPa and 257.53 MPa. As noticed, the spread of Young’s modulus was relatively large. The upper value was approximately three times higher than the lower value of the above-mentioned compartment. With regards to the investigation of the thermal conductivity of biomass-based briquettes, Sova et al. [3] conducted a theoretical analysis of the thermal conductivity of wood cells. They developed equations describing the effective transverse thermal conductivity. They also confirmed their modeling with experimental measurements. Rbihi et al. [4] determined thermal conductivity for cellulose. It was 0.0339 W/(K∙m) at the density of 0.438 kg/m^3^. Wang et al. [5] also conducted research on the thermal conductivity of various biomass-based materials. They obtained thermal conductivity between 0.3 and 0.32 W/(K∙m).

With regard to the durability and stress analysis of the densified biomass, Markowski [6] proposed testing the durability of pellets in accordance with the PN-EN ISO 17831-1 standard. Details of the research methodology proposed in this standard are discussed in another chapter of this article. It should be noted that this standard has become an interpretation of the briquette/pellet durability analysis. Durability tests based on this standard were also carried out by Dyjakon et al. [7] and Lemos [8], who investigated the use of pellets in metallurgical furnaces. They both concluded that the mechanical durability of briquettes decreases with the increase in particle size. Another method of briquette durability measurement was proposed by Bembenek and Hryniewicz [9]. Their proposed methodology was based on the loss of mass caused by dropping a sample from a height of two meters onto a hard surface. The second method proposed by them, necessary for the analysis of the strength properties of the briquette, was compression on a Zwick Roell testing machine. Compressive strength tests of ash briquette samples were also conducted by Borowski [10]. He tested cylindrical samples compressed axially and the so-called Brazilian method, i.e., perpendicular to the cylinder axis, depending on the amount of binder used. Tests of briquette durability and briquette density depending on the compaction temperature in the range of 200–250 °C were carried out by Niedziółka and Szpryngiel [11]. They confirmed that the increase in temperature increased both tested values; however, they did not refer to the direct relationship between durability and density. Chin [12] and O’Dogerty [13] presented a different approach to the analysis of compressed biomass. They analyzed the dependence of the density of the compacted biomass on the compression pressure. According to their results, this dependence is in log correlation. Mechanical strength analysis of briquettes depending on the compaction pressure was also carried out by Nikiforov et al. [14]. They concluded that the compaction pressure of 20 MPa was sufficient for the briquette to have a sufficiently high density and burn stably. The relationship between density and compression pressure was also investigated by Plistil and his team [15]. In this work, they found a positive dependence between the density and the compressive breaking force on the compaction pressure. Research on the compressive strength of the briquette was carried out, among others, by Rahman et al. [16]. However, this work concerned fine coal briquettes and presented only the results of the compression test, without further analysis, including the determination of Young’s modulus and Poisson’s ratio. Kaliyan [17] investigated the efficiency of the compaction process in order to produce a sustainable pellet. He came to the conclusion that parameters, such as strength and durability of compacted products, can be determined by testing the material depending on the binder composition and compaction pressure. According to him, mechanical resistance was defined as compressive and impact resistance, as well as water resistance. In contrast, durability according to Kaliyan was defined as abrasion resistance [17]. In conclusion, the author raised an important problem of developing standards for acceptance criteria for the strength and durability levels of briquettes/pellets. For this purpose, they provided the guidelines needed to develop such standards. Chłopek [18] conducted experimental studies of the process of densification and consolidation of composite fuels, which were mixtures of coal and biomass. He examined a number of models in the field of granulation and compaction theory. Based on his research, it was shown that blends containing traditional solid fuels and biomass can be successfully subjected to pressure agglomeration in a granulator with a flat matrix, assuming the correct selection of process parameters. Roman [19] presented an analysis of the course of biomass compaction by measuring the resistance of the tested briquetted material. He related the resistance to the stresses and deformation of the material pressed on a press with a pressure of up to 100 kN. The work of Rejdak and others is also interesting in this regard [20]. They tested the strength and density of briquetted material from energy willow (Salix Viminalis) depending on the material temperature and compaction pressure during the production process. They used the Brazilian tensile strength method for compressive strength tests. Świętochowski et al. [21] presented the results of their research on the determination of Young’s modulus depending on the deformation method of the compressed material and its diameter. They found that the mean values of the modulus of elasticity and the maximum stresses were in most cases lower for the larger sample sizes (briquettes) than for the pellet size samples. The procedure and methodology similar to that described in ISO 17831 was also applied by Temmerman [22]. He tested the durability defined as briquette abrasion in a drum machine.

In terms of the standards for the durability of briquettes, several standards have been developed that introduce certain methods and formulas for fuel presented as densified biomass in the form of both briquettes and pellets. The ISO 17831 standard has been introduced for biomass which is produced for conversion to heat and/or electricity. The literature review has been carried out with particular emphasis on the research on the durability of compacted biomass. It can be observed that most studies focused on linking durability with biomass density and compaction parameters (pressure and temperature). The concept of the durability of densified biomass has a number of definitions, most of which are reduced to the phenomenon of material abrasiveness and the methods of its measurement. In this case, the vast majority of tests are based on the research methodology described in the ISO 17831 standard. It was considered that the problem of testing the durability and strength properties of compacted biomass was not fully developed. Hence, it was assumed that the research process was properly prepared, and it requires thorough analysis and development of the research methodology, which will contribute to the practical usefulness of the research. Firstly, it was concluded that testing the durability (in many publications called strength) of the briquette using the methodology presented in the ISO 17831 standard will not be adequate to meet expectations regarding the results and conclusions. It does not give results that can be used to find the relationship between density and the compaction parameters (pressure and temperature).

Summing up, the main objective of the research in this article is to determine the relationship between compaction pressure and both density and destructive force at elevated temperatures, and to determine the thermal conductivity of the compacted material. These data are necessary to analyze the compaction and torrefaction processes and for the optimization of these processes. Moreover, it was found that the results of the so-called durability/strength tests with the methodology outlined in ISO 17831 cannot be used to analyze the actual material properties of biomass compacted into briquettes. The tests presented in the article were carried out on Virginia mallow (Sida Hermaphrodita) feed material. It is an energetic crop with the potential for combustion. It is feedstock for biogas fermentation plants as well as fodder for cattle and other domestic animals [23,24].

Finally, let us emphasize that our goal in this article was not only to show the properties of Virginia mallow towards compaction and in the further the torrefaction process, but also to show the methodology and some trends appearing between properties for briquetted material. It can be concluded that based on this proposed research methodology, other energy crops and/or waste biomass can be analyzed in terms of their effective compaction.

## 2. Materials and Methods

### 2.1. Rod Sample Preparation

The samples used for tests were characterized by the following dimensions:diameter: 25 mmlength: 100 mm.

The input material for the production of samples were plants of the Virginia mallow (Sida hermaphrodita) crop presented in the form of chips with the size of 0.5 to 5 mm compacted at various pressures of 100, 200, 400, 600, 800, and 1000 bar and at a temperature of 20 and 200 °C. For the measurements of thermal conductivity, the briquetting parameters were as follows: compaction pressure 200 bar and temperature 200 °C.

### 2.2. Methodology for Thermal Conductivity Determination

The thermal conductivity was determined on the basis of measurements and modeling of heat conduction at a transient flow inside the sample, assuming that the heated sample was an infinitely long cylinder located horizontally in an oven, taking this into account in modeling the heat flow through it. This results in a forced airflow, which ensures the Biot number is high enough to consider mainly heat conduction, rather than convection, as the leading type of heat transfer during the experiment. The Biot number expresses the ratio of heat conduction resistance through the solid mass to heat convection resistance between the body and the environment. The Biot number has been estimated to be approximately 9.

Simplifications:The heat flux *q”* (W/m^2^) was calculated from the conductivity Equation (1) assuming the temperature gradient inside the rod close to the rod surface
(1)$$q^{\prime \prime }=k_{mean}\cdot \frac{dT}{dr},$$The heat flux $q^{\prime \prime }$ was constant across the pellet*k_mean_* is the material conductivity determined by Equation (2), calculated for the temperature gradient inside the rod close to the rod surface
(2)$$k_{mean}=\frac{k\left(T_{s-1}\right)+k\left(T_{s}\right)}{2},$$*T_s_* represents the temperature taken from the endpoint near the surface of the rod at quasi-steady-state conditions. This means conditions under which the temperature has been stabilized, whereas *T_s_*_−1_ is the temperature inside the rod at a distance of a single step in the discrete meshThe contact resistance between the thermocouple and the material surface has been ignored

It was assumed that the cylinder was infinitely long and the axis was symmetrical, which ensures 1D heat conduction. Temperature *T* is a function of 2 variables: time *t* and radial coordinate *r*, *T* = *T*(*t*, *r*).

The temperature determination can be obtained by solving Equation (3) for heat conduction in a cylindrical coordinate system (*r*, θ, *z*).
(3)$$\frac{1}{r}\frac{d}{dr}\left(r\cdot k\frac{dT}{dr}\right)+\frac{1}{r^{2}}\frac{d}{d\theta }\left(k\frac{dT}{d\theta }\right)+\frac{d}{dz}\left(k\frac{dT}{dz}\right)+{\dot{e}}_{gen}=\rho c\frac{dT}{dt},$$
where: *k* represents the thermal conductivity coefficient; ρ represents density; and *c* represents specific heat.

From the general form of heat conduction (Equation (3)), a simplified form can be obtained that applies to the experiment case. Assuming there is no heat generation inside the rod body, homogeneous properties (*k*, ρ, and *c*), and the fact that the temperature distribution depends only on the radial coordinate (*r*), Equation (3) can be written as follows (Equation (4)):(4)$$\frac{d^{2}T}{dr^{2}}+\frac{1}{r}\frac{dT}{dr}=\frac{1}{\alpha }\frac{dT}{dt},$$
where $\alpha =\frac{k}{\rho c}$ represents thermal diffusivity ((*k* represents the thermal conductivity coefficient; ρ represents density; and *c* represents specific heat).

They have been adopted as follows:
ρ = 750 kg/m^3^

c = 1650 J/(kg∙K).

The parameter k was taken as a parameter varying from 0.03 to 0.08 in this study. The problem is considered to be axially symmetric, hence the temperature distribution neither depends on z nor θ coordinates.

As depicted in Figure 1, the Dirichlet boundary conditions related to the symmetry of the cylinder and external rod surface temperature are as follows (Equations (5)–(7)):Surface temperature changing with time (measured by the thermocouple surface):
(5)$$T_{1}=T_{surface}=T_{measured}\;at\;r=r_{max},$$
(6)$$T_{2}=T_{center}=T_{measured}\;at\;r=0,$$The initial conditions are as follows:
(7)$$T\left(r,t\right)=T_{initial}\;at\;t=0\;and\;r=r_{max},$$

Additionally, the calculations took into account the Neumann boundary condition concerning the temperature change inside the bar center (Equation (8)).
(8)$$\frac{dT}{dr}=0\;at\;r=0$$

Inside the sample (rod), a hole with a diameter of 2 + 0.2 mm and a depth of 75 mm was drilled. Then, a conductive paste was injected into the drilled hole. This was undertaken to reduce the thermal contact resistance between the material and the thermocouple. The thermocouple with a diameter of 2 mm was inserted into the drilled hole. The thermocouple wiring system was insulated to minimize heat conduction through the thermocouple itself. In order to measure the rod surface temperature, the thermocouples were mounted on the external rod surface (Figure 2). The heating and cooling experiments were conducted for the top configuration (surface thermocouple on the top of the rod) and the bottom configuration (surface thermocouple on the bottom of the rod). The tests were repeated 4 times. Figure 2 shows the placement of the thermocouples on the sample (rod) in the top and bottom configurations.

#### 2.2.1. Test Procedure

The oven was set to a temperature in the range of 100 to 110 °C. After the oven temperature had stabilized, the rod specimen was placed inside the oven. The heating experiment was carried out for a time that allowed the temperature inside the rod to stabilize. The temperature data were collected continuously by an analog-to-digital converter coupled with a data logger and a computer. In Figure 2, the schematic diagram of the constructed system for the heating experiment is presented. The cooling experiment was conducted outside the oven. The heated rod was removed from the oven and hung on threads, maintaining the same configuration as in the heating experiment. The rod was subjected to free air convection and ambient temperature conditions to cool the rod sample.

#### 2.2.2. Heating and Cooling Tests

The normalized temperature was introduced to make a comparison between tests which can vary with respect to the final temperature.

The normalized temperature for heating tests was calculated using Equation (9):(9)$$\theta_{h}=\frac{T\left(r,t\right)-T_{initial}}{T_{\infty }-T_{initial}},$$
where:
*T_initial_* represents the temperature for *t* = 0,*T*∞ represents the final temperature after the rod has reached steady-state conditions.

The normalized temperature for the cooling tests was calculated as follows (Equation (10)):(10)$$\theta_{c}=\frac{T\left(r,t\right)-T_{\infty }}{T_{initial}-T_{\infty }},$$
where:
*T_initial_* represents the temperature for time *t* = 0,*T*_∞_ represents the temperature after the rod has reached steady-state conditions.

Hence, Equation (4) remains the same form after temperature normalization (Equation (11)):(11)$$\frac{d^{2}\theta }{dr^{2}}+\frac{1}{r}\frac{d\theta }{dr}=\frac{1}{\alpha }\frac{d\theta }{dt},$$

With regard to the steel coating of the thermocouple and its potential impact on measurement accuracy, it was found that the average thermal conductivity of stainless steel is in the range of 13–17 W/(K∙m). This is several orders of magnitude higher than the material conductivity. It can therefore be concluded that the impact of thermocouple conductivity is almost negligible, and thus can be ignored.

### 2.3. Methodology for Stress Analysis

Experimental tests were carried out in the field of material compressive strength, where the basic measured quantities were:force acting on the samplesample deformation

Tensile tests were not carried out. This was due to the high brittleness of the tested material and difficulties in the method of effective fastening, which had no effect on the tensile test.

For the experimental tests, samples in the form of cylindrical briquettes pressed at various pressures and temperatures were used. A Zwick Roell Z100 testing machine was used to determine the correlations between compaction pressure and material density, as well as the maximum destructive force. The pressure of 200 bar was adopted as the most economically viable in the briquetting technology.

In terms of the compaction temperature, 2 temperatures were adopted:200 ± 1 °Cambient temperature (approximately 20 ± 0.5 °C)

The justification for the selection of the compaction temperature of 200°C results directly from the analysis of the target application of the torrefaction briquetting technology on an industrial scale.

### 2.4. Compressive Strength Test

After briquetting, the samples were tested for compressive strength in a plane perpendicular to the symmetry axis (Figure 3a) of the sample cylinder and in a plane perpendicular to it (Figure 3b).

Investigation on the correlation between the density of the compacted biomass and the compaction pressure was carried out. For this purpose, the displacement of the press punch (piston) was measured as a function of the compaction force exerted by the punch on the material to be compacted.

The deformation ε as a result of compaction was calculated using Equation (12):(12)$$\varepsilon =\frac{\Delta h}{h_{p}}\cdot 100\%$$
where:
Δh represents the change in height of the sample during compaction under a given pressure,h_p_ represents the initial sample height before compaction.

The initial sample height h_p_ was assumed to be the specimen height after initial punch pressure at 5 bar. The deformation calculated in this way was used to investigate the relationship with the compaction pressure. Due to the expected logarithmic relationship, the compression pressure and strain graphs are plotted on a logarithmic scale as a function of the compression pressure.

## 3. Results and Discussion

### 3.1. Results from Conductivity Tests

Figure 4 presents the temperature data from exemplary tests for heating and cooling cases.

As can be seen in Figure 4, the temperature T2 inside the sample varies typically to the theoretically described heat transfer to or from the sample and asymptotically reaches the final temperature. However, to recognize the effect of cooling and heating it is recommended to plot the normalized temperature versus time to compare all data sets as shown in Figure 5. From the normalized temperature diagrams, it can be concluded that the heat transfer by conduction in the heating tests was higher in comparison to the cooling tests. Of course, it can be riposted that the heat transfer by convection at the surface of the sample was higher in the heating test compared to the cooling test, although, both these cooling and heating tests were performed in an environment without forced air (no turbulence) near the surface of the sample. Hence, this is another observation of this study.

Figure 6 shows an example of the temperature distribution T2 inside the sample as a function of time during the cooling test. In the figure, the inner temperature T2 from the tests was compared with the inner temperature calculated from Equation (4) (marked as T2 calc) for the three various conductivity coefficients: 0.36, 0.48, and 0.60 W/(K∙m).

Based on this procedure, the conductivity coefficient k was determined for the four cooling and the four heating tests. The average conductivity k was determined as follows:for cooling tests: k = 0.048 ± 0.001 W/(K∙m)for heating tests: k = 0.049 ± 0.003 W/(K∙m)

As observed, the results from the cooling tests seemed to be more reliable due to the lower uncertainty of 0.001 W/(K∙m), hence, for further analysis, the cooling tests were proposed as more reliable in this case.

### 3.2. Stress Analysis

Figure 7a shows exemplary compression test results of samples made of briquettes compacted under various pressures from 100 to 1000 bar at a temperature of 200 °C in the axis of symmetry of a cylindrical sample (as shown in Figure 3a). The results of deformation from the performed strength test as a function of load (Figure 3b) were not taken into account, because the obtained results were burdened with a high measurement uncertainty resulting from a large discrepancy in the anisotropy of the compacted samples. Moreover, in this case, the alignment of the samples on the machine had a significant influence on the results. In the graphs (Figure 7a), the maximum compressive/destructive force is marked with a circle. After reaching this value, the compressed material was destroyed as a result of its scattering. Based on the observation of the destruction of the sample, it could be noticed that the samples did not show any plasticity properties and the first visual signs of sample crushing occurred when the destructive force reached its maximum. Hence, it can be concluded that the destruction of the internal structure of the sample probably occurred a little earlier, just before reaching the maximum destructive force. This effect was clearly visible for the sample compressed at a pressure of 100 bar. Figure 7b shows the maximum destructive force as a function of compaction pressure for the briquettes presented in Figure 7a. As can be seen, there is a linear correlation between the maximum destructive force and the compaction pressure. This observation can be used in further analysis of abrasion and compacted briquettes. These tests were repeated three times. Based on these tests, mean values were determined. However, the plots in Figure 7, on the other hand, are plots from individual compression tests shown as an example to confirm the trend between the compaction pressure and the maximum destructive force. On the other hand, averaging these runs vs. the press punch displacement led to a large discrepancy regarding clearance removal at the starting point of each compression test.

According to Equation (12), the deformation in relation to the compaction pressure is presented in Figure 8. As can be observed, the logarithmic correlation appears when the compaction pressure does not exceed 350 bar, which causes the deformation of the briquetted samples of 55 and 60% depending on compaction temperature of 20 and 200 °C, respectively.

In terms of the relationship between the density of the compacted briquette sample and the compaction pressure, the thesis of Chin and O’Dogerty confirmed that the logarithmic dependence (Equation (13)) best fits the experimental results, linking the density with the compaction pressure [12,13]. However, the values of the a and b coefficients, which significantly exceeded the ranges proposed by these scientists, have not been confirmed. The dependence of the density as a function of compaction pressure with trend lines and their equations is shown in Figure 9 and in Table 1. Chin and O’Dogerty proposed to present this relationship in an analytical form using Equation (13) [12,13]:(13)$$\rho =a\cdot \ln\left(p\right)+b\;\left(kg/m^{3}\right)$$
where *a* and *b* are empirically determined coefficients.

For the graphs presented in Figure 9a, the trend lines were determined in accordance with the logarithmic relationship and the coefficients a and b for Equation (13) were calculated. The regression coefficient (R2) for all cases is very high, almost close to one, which may indicate a very good correlation of the experimental results with the logarithmic function presented in Table 1.

Moreover, it can be seen in Figure 9 that the density of the compacted biomass also depends on the temperature. The change of the compaction temperature from 20 to 200 °C significantly increased the density by approximately 300 kg/m^3^ at a compaction pressure of 200 bar. Moreover, it can be seen that the compaction process at a pressure higher than 400 bar is not economically feasible since the density no longer increases as much as it does with an increase in pressure in the range of up to 400 bar. One can conclude from Figure 9b that the increase in density is by 15% (from 530 to 610 kg/m^3^ at 20 °C), whereas the compaction pressure had to increase twice from 400 to 800 bar. Furthermore, we can suggest the optimal compaction pressure due to energy consumption is 200 bar. By comparing the results of the briquette’s density at 450 kg/m^3^ obtained under 200 bar with a density of 610 kg/m^3^ at 800 bar compaction pressure, a density increase of 35% can be observed. However, the energy required for briquetting at 800 bar is much higher and the briquetting process becomes less economical. Moreover, the destructive force is also high enough for the compaction pressure of 200 bar that a crushing or abrasion will not remarkably deteriorate the compacted material. Therefore, it was considered inadvisable to increase the compaction pressure, which further increases the cost of compaction due to the need for more energy.

## 4. Conclusions

The following conclusions were drawn from the research work:The average thermal conductivity was calculated from the equation of heat conduction in the unsteady state for cylindrical coordinates. The obtained thermal conductivity was 0.048 ± 0.001 W/(K∙m). As mentioned in Section 1, thermal conductivity determined by others was between 0.030 and 0.039 W/(K∙m) for various biomasses densified at various pressures.The cooling experiment for thermal conductivity determination allowed for a more precise adjustment of the theoretical temperature to the experimental data. This can be caused by the lower impact of the thermocouple wiring system on heat conduction and additionally, the impact of the heat convection may be marginal.The proposed methodology for determining the thermal conductivity for biomass-based briquettes can be considered a useful tool for other low-conductivity materials.The destructive force is strictly in line with the compaction pressure.The temperature of the compaction process of 200 °C can be considered optimal regarding direct briquetting after the torrefaction process has taken place at a typical temperature range between 200 and 260 °C.The compaction pressure of 200 bar has been proposed as an optimal compaction pressure due to economic considerations related to the energy required for the compaction process.

## Figures and Tables

**Figure 1 materials-15-08458-f001:**
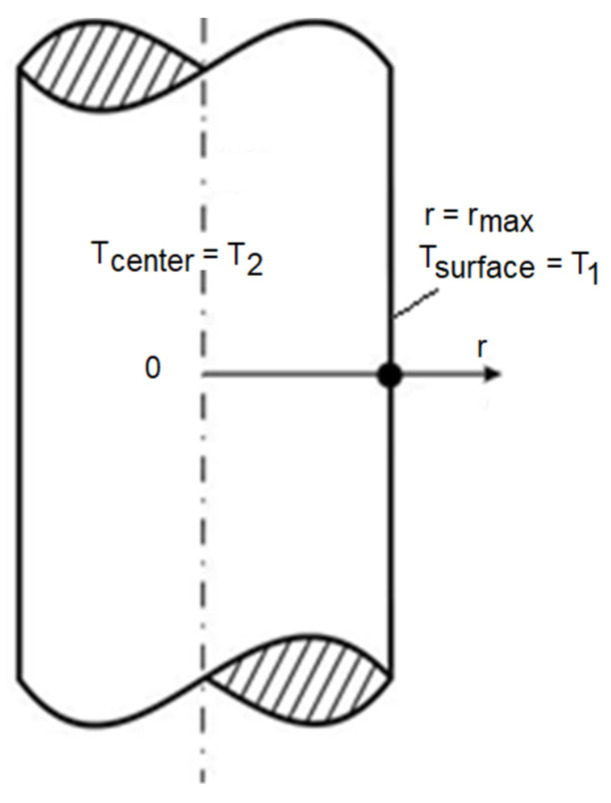
Temperature distribution inside an infinite rod.

**Figure 2 materials-15-08458-f002:**
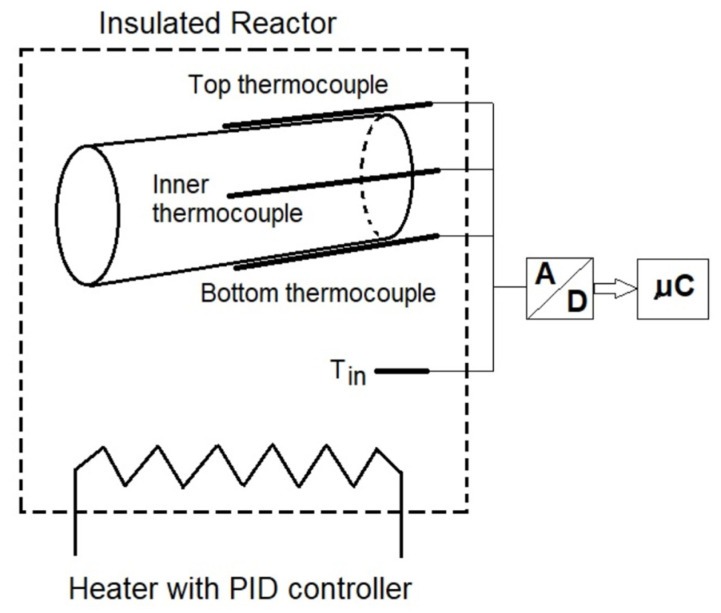
The location of thermocouples in the sample.

**Figure 3 materials-15-08458-f003:**
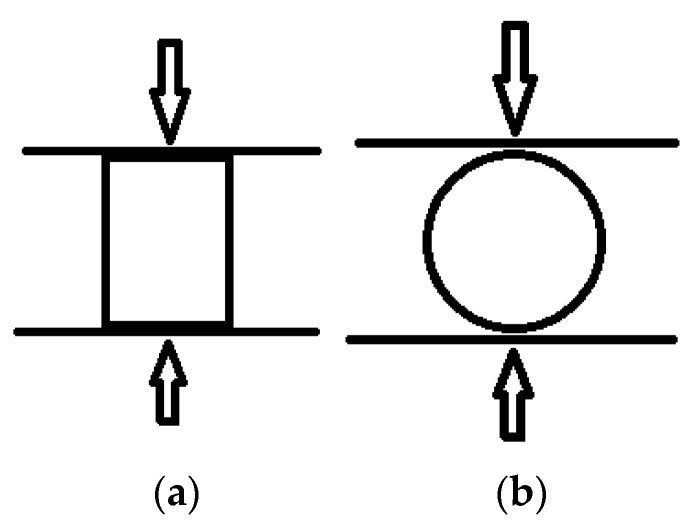
Sample compressive strength test scheme: (**a**) in a plane perpendicular to the axis of symmetry; (**b**) in a plane perpendicular to it.

**Figure 4 materials-15-08458-f004:**
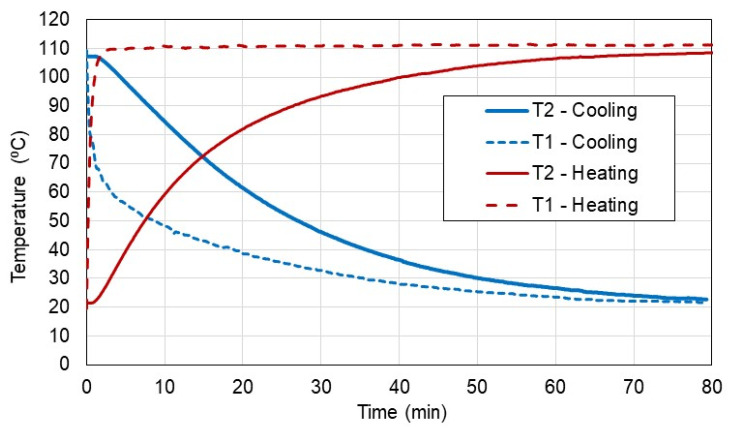
Inner T2 and outer T1 temperatures for heating and cooling tests.

**Figure 5 materials-15-08458-f005:**
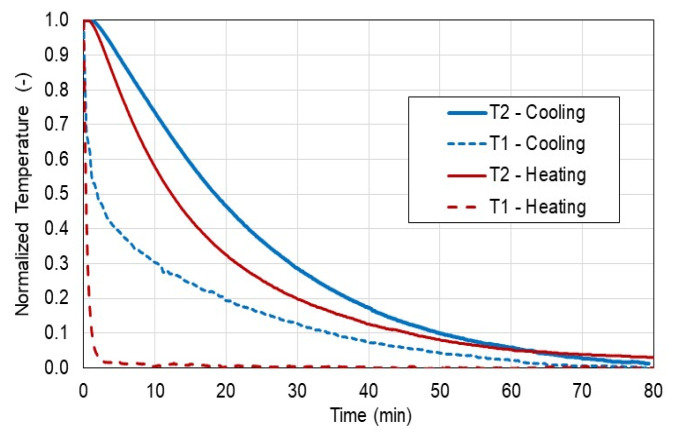
Normalized temperature for heating and cooling tests.

**Figure 6 materials-15-08458-f006:**
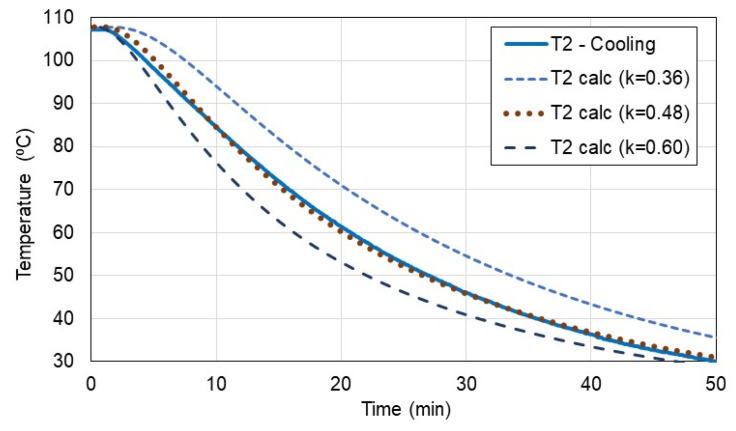
Comparison of inner temperature from experiment and calculations.

**Figure 7 materials-15-08458-f007:**
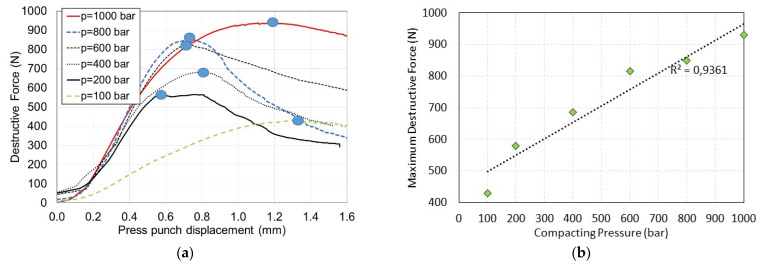
(**a**) The destructive force vs. press punch displacement for tests with various compaction pressures for briquettes at the temperature of 200 °C; (**b**) maximum destructive force vs. compaction pressure for briquettes.

**Figure 8 materials-15-08458-f008:**
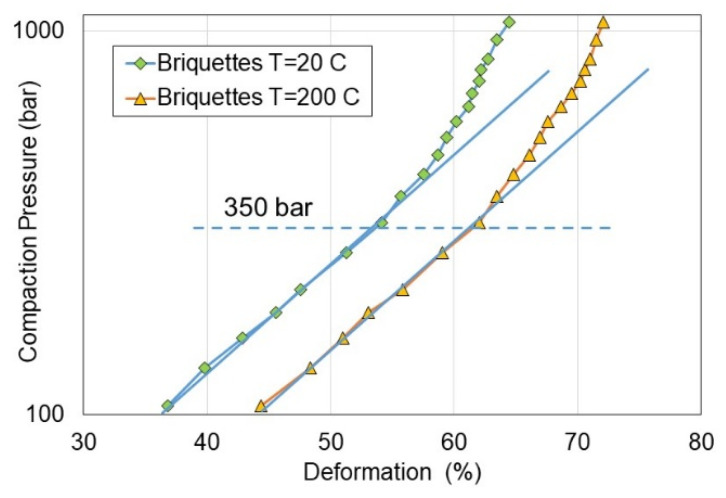
Compaction pressure course vs. deformation of briquettes.

**Figure 9 materials-15-08458-f009:**
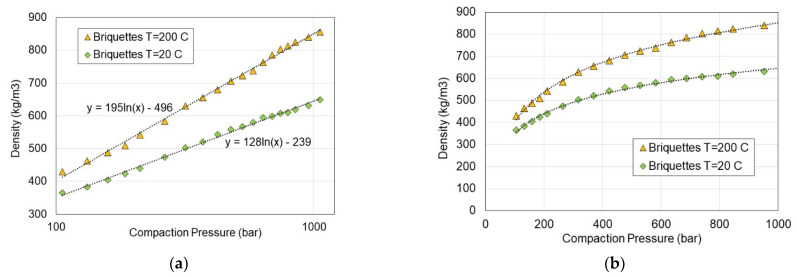
Density of compacted biomass versus pressure and temperature of compaction (**a**) in log scale; (**b**) in linear scale.

**Table 1 materials-15-08458-t001:** Coefficients of Equation (13) and the regression coefficient R^2^.

	Coefficient a	Coefficient b	R^2^
Briquettes at T = 200 °C	195	−496	0.997
Briquettes at T = 20 °C	128	−239	0.997

## Data Availability

Not applicable.

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
