# Peer review of "Thermal and Stress Properties of Briquettes from Virginia Mallow Energetic Crops"

_materials, 2022, doi:10.3390/ma15238458_

Round 1

Reviewer 1 Report

The aim of the submitted manuscript article "Thermal and stress properties of briquettes from Virginia mallow crops" is to determine the relationship between compaction pressure and density and the breaking force at elevated temperatures as well as to determine the thermal conductivity of the compacted material based on samples whose starting material was Virginia mallow crops in the form of chips up to 5 mm pressed under different pressure (100 - 1000 bar) and at 20 and 200°C.  

The relevance of the study is due to several factors. First, energy production using renewable resources, including the combustion of biofuels, is of some practical interest even now. Thus, agricultural or forest products, or their waste, can be used to produce biochar. From this point of view, a second important factor is the enormous resources of plant materials (e.g. wood and its processing waste, straw, agricultural waste, etc.) that are produced every year. This renewable raw material requires an integrated approach to its use, i.e., safe and economically viable recycling.

As the reviewer studied the text of the submitted manuscript, a number of questions/recommendations arose: 

1. the literature review should be structured. It is necessary to make 2-3 introductory sentences on the use of plant raw materials for the production of briquettes/pellets. When citing literary sources, authors often resort to stating the facts of any experiments without indicating the results obtained. Or, on the contrary, they overload the text with an excessive number of figures, even giving an equation, which makes it difficult to perceive the text and understand the overall picture of research in this field. At the same time, in the section on discussion of results, the reviewer just recommends to put specific figures, allowing to characterize and compare the results obtained by the authors with the results of other authors (for example, on thermal conductivity).

I ask to pay attention to functional properties of raw materials received from waste products: 10.1016/j.net.2022.04.005, 10.1134/S1066362219020097 

2. The selected species of Virginia mallow plant (Sida hermaphrodita) used to produce briquettes is not substantiated. If there is any data in the literature on the use of such a species, on approximate volumes (this type of plant is considered to be yielding), should be cited, at least at the level of references. 

3. In the section "Materials and methods" the authors give the conclusion of equations, which makes the experimental part equal to almost 40% of the total volume of the article. If possible, revise the structure and put the derivation of equations in the results and discussion.

4. Specify the country of manufacture of the test machine (Zwick Roell Z100).

5. What is the reason for the choice of pressure (200 bar) in the briquetting technology? If the authors did not directly conduct experiments on pressure selection, then this conclusion should be removed from the "Conclusions" section.

6. The conclusions need to be supplemented and specified. As presented, the reviewer has the impression that they are weakly related to the article. It is also necessary to make a conclusion about the practical importance of the research conducted. This will be easier to do if in the section "Results and discussion" to give a comparative analysis of the parameters of briquettes obtained by different methods from vegetable raw materials.

7. The abstract requires more specificity concerning the work itself. It is necessary to remove the sentences with the facts of a general nature and emphasize the own results.

Reviewer 2 Report

Please, include the number of repetitions that you made, especially in the stress analysis.  

Reviewer 3 Report

1. Include about the feedstocks in the introduction section.

2.Try to include all the results in the conclusion.

3. Try to find some other method to calculate the stress analysis

Round 2

Reviewer 1 Report

The manuscript in its current form can be published

Author Response

Please see the file attached.
